# DisCo: Improving Compositional Generalization in Visual Reasoning through Distribution Coverage

**Joy Hsu**                                                                 *joycj@stanford.edu*
*Department of Computer Science, Stanford University*

**Jiayuan Mao**                                                            *jiayuanm@mit.edu*
*Department of Electrical Engineering and Computer Science, Massachusetts Institute of Technology*

**Jiajun Wu**                                                           *jiajunwu@cs.stanford.edu*
*Department of Computer Science, Stanford University*

**Reviewed on OpenReview:** *https://openreview.net/forum?id=EgHnKOLaKW*

## Abstract

We present DisCo, a learning paradigm for improving compositional generalization of visual reasoning models by leveraging *unlabeled, out-of-distribution* images from the test distribution. DisCo has two components. The first is an iterative pseudo-labeling framework with an entropy measure, which effectively labels images of novel attribute compositions paired with randomly sampled questions. The second is a distribution coverage metric, serving as a model selection strategy that approximates generalization capability to test examples drawn from a different attribute combination distribution to the train set, without the use of labeled data from the test distribution. Both components are built on strong empirical evidence of the correlation between the chosen metric and model generalization, and improve distribution coverage on unlabeled images. We apply DisCo to visual question answering, with three backbone networks (FiLM, TbD-net, and the Neuro-Symbolic Concept Learner), and demonstrate that it consistently enhances performance on a variety of compositional generalization tasks with varying levels of train data bias.

## 1 Introduction

A long-standing goal of visual reasoning is to build machines that can respond to queries about images in a flexible and general way as humans do. To achieve this, machines must contend with the combinatorial complexity of natural images and queries: a scene has multiple objects, each object has a collection of attributes, and objects form various spatial and functional relationships. The combinatorial explosion of image spaces, together with practical data limitation in downstream tasks, makes many learning problems ill-posed (Bienenstock et al., 1996; Lake et al., 2017). In this paper, we focus on the compositional generalization to novel combinations of object attributes, generalizing from the reasoning of *blue cubes* and *red cylinders* to that of *red cubes*. This is an important desideratum for machine learning systems: it is impossible for any dataset to include all possible combinations of object attributes for model training.

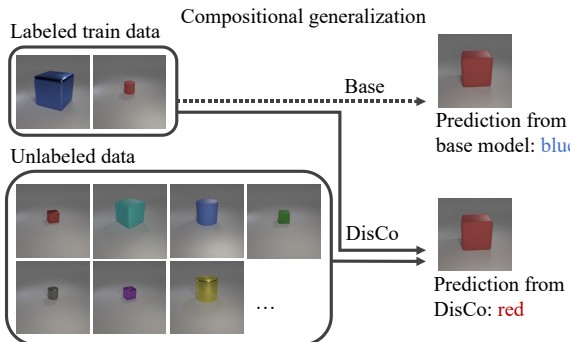

Figure 1: The base VQA model in its original training paradigm is trained on labeled data, while DisCo leverages *unlabeled, out-of-distribution* images from the test distribution to improve compositional generalization performance.

We address this problem by introducing DisCo, a learning paradigm that leverages *unlabeled, out-of-distribution* images from the test distribution to help visual reasoning systems better generalize compositionally (See Figure 1). Concretely, we focus on the task of visual question answering (VQA), though our framework is model-agnostic and can be used for a variety of vision domains that have combinatorial structures, such as in the tasks of referring expression comprehension, grounded instructions, and robotic manipulation (Achlioptas et al., 2020; Shridhar et al., 2020; Chevalier-Boisvert et al., 2019; Shridhar et al., 2022). Given a labeled train set of image, question, and answer triplets, and unlabeled images from the test distribution that contains novel attribute combinations, DisCo bootstraps a visual reasoning model by iteratively mining data instances derived from the unlabeled dataset. Starting from a base VQA model trained on a labeled, possibly biased dataset, our framework couples unlabeled images that are out-of-distribution with randomly sampled questions, and discovers pairs that are answerable. These newly-created data points are trained with equal weighting as the original labeled data points, and bootstrap learning: the model gradually labels more and more new image-question pairs, increasing *distribution coverage* on out-of-distribution image sets that contain novel attribute compositions, with individual attributes seen with labeled data but under different combinations.

Pseudo-labeling is particularly difficult in the visual question answering setting, as given an unlabeled question-image pair, 1) there is a high probability of *presupposition* adherence failure (e.g., the question asks about the color of the cube in the image, but there is no cube), and 2) the current model may not have the capability to reason about the out-of-distribution image correctly. These two failure cases make pseudo-labeling images from the test distribution especially noisy. The effectiveness of our iterative learning paradigm is hence based on the empirical insight that, for a pretrained visual reasoning model, the *entropy* of its predicted answer distribution correlates strongly with its accuracy on images with novel attribute compositions. This entropy metric approximates both presupposition adherence as well as compositional generalization accuracy, thus is crucial to DisCo choosing unlabeled samples to be pseudo-labeled.

Moreover, in the compositional generalization setting, model tuning and model selection are challenging, as there is limited access to labeled data in the test distribution. Validation sets are in the same distribution as the train set, and hence unable to approximate test set performance, leaving methods unable to select for model checkpoints that best generalize to unseen data. To address this issue, we propose a *distribution coverage* metric, which computes the percentage of unlabeled, out-of-distribution images drawn from the test distribution that can be answered confidently by the current VQA model. This distribution coverage measure well approximates model accuracy on generalization test splits with unseen attribute compositions, allowing us to effectively tune and select models without labeled data points from the test distribution.

We validate the effectiveness of our approach on biased versions of the CLEVR dataset (Johnson et al., 2017) created for compositional generalization in visual reasoning. Specifically, in addition to the original CLEVR CoGenT dataset, we also construct datasets that contain questions with referred objects as well as one-hop relational questions, and demonstrate generalization improvement with DisCo compared to base VQA models with their original training paradigm. We demonstrate that DisCo consistently helps three VQA models, FiLM (Perez et al., 2018), TbD-net (Mascharka et al., 2018), and the Neuro-Symbolic Concept Learner (NS-CL; Mao et al., 2019), perform better on a test set with a distribution shift of attribute combinations from the train set. We also show that our framework outperforms other methods that leverage unlabeled data, including generative modeling and contrastive learning, on different levels of biases in training data. Our model exhibits an advantage in generalization to novel attribute combinations, and is a step towards contending with the combinatorial complexity of the visual world.

## 2 Related work

**Compositional generalization.** Prior work on improving the compositional generalization of visual reasoning systems generally falls into two groups. The first line of research leverages explicit structures of compositional concepts, such as visual grammars (Zhu & Mumford, 2007; Chen et al., 2007), compositional embeddings (Misra et al., 2017), neural operators (Nagarajan & Grauman, 2018), neural module networks (Purushwalkam et al., 2019), and causal graphs (Niu et al., 2021; Yang et al., 2021b). The second line of research introduces additional supervision, such as the taxonomy of concepts to improve model generaliza-

tion (Han et al., 2019). We present a novel perspective on compositional generalization, which is to leverage *unlabeled, out-of-distribution* data from the test distribution.

**Self-supervised learning for visual reasoning.** Our iterative pseudo-labeling framework is also related to prior work on self-supervised learning for visual reasoning. Specifically, Kim et al. (2021) and Lin & Parikh (2017) use active learning to select image and question pairs to be labeled. Askarian et al. (2021) and Li et al. (2020) use curriculum learning to prioritize training data for visual reasoning models. Methods such as Kim et al. (2019); Zhu et al. (2020); Liu et al. (2018) apply adversarial self-supervised learning to overcome language priors in vision-language models. Although some earlier work has explored similar entropy-based measures as ours, our work differs from them in two key aspects. First, our method studies a different setting, where no additional labels will be requested on the unlabeled dataset. Second, in contrast to data efficiency or task performance, our work shows that an entropy-based measure is especially beneficial for the compositional generalization capability of models in a test set with novel attribute combinations.

**Semi-supervised learning.** DisCo generally falls into the category of semi-supervised learning, whose idea is to leverage unlabeled data to improve model performance. Specifically, early work on pseudo-labeling (Nigam & Ghani, 2000; Grandvalet & Bengio, 2004) has drawn important theoretical connections between entropy-based self-training and expectation maximization algorithms. Prior work has also used signals such as high values in density-based clustering (Choi et al., 2019) and label propagation (Iscen et al., 2019) to choose and infer pseudo-labeled examples, and introduced regularization techniques to learn better between-class separability (Shi et al., 2018). A related work, Rizve et al. (2021), proposes choosing pseudo-labels based on confidence and uncertainty of network predictions for classification, while DisCo tackles the complex VQA task with out-of-distribution images from the specified test distribution that contain unseen attribute combinations. We refer readers to Van Engelen & Hoos (2020) and Yang et al. (2021a) as two recent comprehensive surveys. Our paper uses a similar broader framework and focuses on the empirical evidence in visual reasoning domains to use unlabeled images to generalize in combinatorially complex settings.

## 3 Methods

We present DisCo as a method to leverage *unlabeled, out-of-distribution* image data from the test distribution for compositional generalization. At a high level, DisCo is a pseudo-labeling framework applied to VQA models, that iteratively learns to label images farther from the train data distribution. Intuitively, DisCo chooses unlabeled images from a test distribution of novel attribute compositions that the current reasoning model can effectively answer. As training progresses, DisCo selects more difficult question-image pairs, which increases distribution coverage on the unlabeled image set. We show that in the CLEVR dataset, after training, models can reason about simulated objects in scenes with new attribute compositions, not seen with labels during training.

In this section, we first describe our problem formulation (Section 3.1) and broader learning paradigm with unlabeled images (Section 3.2). We then discuss critical components in the framework. We describe image proposal methods for efficiently generating answerable images in DisCo (Section 3.3). Then, we propose an entropy-based threshold as a measure for accurate pseudo-labeling of out-of-distribution images from the test distribution (Section 3.4). Finally, we present a distribution coverage measure as an effective model selection strategy for compositional generalization (Section 3.5).

### 3.1 Problem formulation

In this paper, we focus on improving the object-level compositional generalization of visual reasoning models. Intuitively, objects in images are associated with many concepts, such as color, shape, and material. During training, the model may only see a finite number of possible concept combinations for objects. The goal of object-level compositional generalization is to let the model trained with limited labeled data generalize to novel object concept compositions. Our proposal is to leverage unlabeled, out-of-distribution images from the test distribution, which are significantly more available than human-annotated data, to improve the performance of visual reasoning models in combinatorially complex domains.

---

**Algorithm 1** The DisCo framework described in Section 3.2.

---

**Input:** $D_{\text{train}}$: the labeled train dataset; $f$: image proposal function derived from the unlabeled test dataset $D_{\text{test}}$; $\mathcal{M}(v, q; \theta)$: visual question answering model; $n$: entropy threshold.

**Output:** $\theta$: $\mathcal{M}(v, q; \theta)$.

1: Pretrain $\mathcal{M}$ with $D_{\text{train}}$.
2: Track distribution coverage in $c$.
3: **for** $i \leftarrow 0$ **do**
4:     $(v_+, q_+, a_+) \sim D_{\text{train}}$
5:     $v_- \leftarrow f()$                           ▷ Image proposal function $D_{\text{test}}.\text{sample}()$ or $GAN.\text{generate}()$, see Section 3.3.
6:     $p_+ \leftarrow \mathcal{M}(v_+, q_+; \theta)$                           ▷ Retrieve predictions from pretrained $\mathcal{M}$.
7:     $p_- \leftarrow \mathcal{M}(v_-, q_+; \theta)$
8:     **if** $\text{entropy}(p_-) < n$ **then**        ▷ Entropy measure to select for pseudo-label targets, see Section 3.4.
9:         Update $c$ with distribution coverage.
10:         Update $\mathcal{M}$ with $xent(p_+, a_+)$.
11:         Update $\mathcal{M}$ with $xent(p_-, \arg\max(p_-))$.                ▷ Update $\mathcal{M}$ with equal weighting of $+$ and $-$.
12:     **else**
13:         Reject sample.
14:     **end if**
15: **end for**
16: Choose checkpoint of $\mathcal{M}$ through $c$.                ▷ Coverage used for model selection, see Section 3.5.

---

DisCo is trained on a labeled dataset of VQA triplets, with each data point containing a visual scene, question, and answer; we denote this as $(v_+, q_+, a_+) \in D_{\text{train}}$. The training dataset only contains a subset of attribute combinations of *colors* and *shapes* (e.g., *blue cubes* but not *red cubes*), while the test dataset contains objects of different color-shape combinations. Our training objective is thus to bootstrap from a VQA model to iteratively improve the distribution coverage of out-of-distribution test examples.

### 3.2 Iterative pseudo-labeling

Our learning paradigm has three steps. First, we assume a base visual question answering model $\mathcal{M}(v, q; \theta)$, and train $\mathcal{M}$ to convergence on $D_{\text{train}}$ without modifications to the original training procedure. Second, we bootstrap the model on *unlabeled, out-of-distribution* data from the test distribution with a proposed pseudo-labeling framework. Lastly, with the distribution coverage produced by DisCo, we perform model selection for a checkpoint of $\mathcal{M}$ that generalizes best to novel attribute compositions. We describe this paradigm in Alg 1 and show the overview in Figure 2.

DisCo utilizes unlabeled images in $D_{\text{test}}$ through an image proposal function $f$ derived from $D_{\text{test}}$ (Alg 1, L5). The proposal function can be a random sampler of unlabeled images in $D_{\text{test}}$, or a learned generative model, such as a generative adversarial network (GAN; Goodfellow et al., 2014), trained on the unlabeled images in $D_{\text{test}}$. Either $f$ yields unlabeled images $v_-$ from the test distribution to be used in our framework.

In the iterative pseudo-labeling process, DisCo couples images $v_-$ with randomly sampled questions $q_+$ from the labeled train set (Alg 1, L7); let $p_-$ denote the answer distribution produced by $\mathcal{M}$, i.e., $p_- = \mathcal{M}(v_-, q_+; \theta)$. We use an entropy measure to select confident predictions such that question-image pairs satisfy presuppositions and are answerable (Alg 1, L8). The $(v_-, q_+)$ pair will be pseudo-labeled with $\arg\max(p_-)$, its own sharpened predictions (Alg 1, L11). During training, we keep track of the percentage of pairs $(v_-, v_+)$ that can be confidently answered by the model (Alg 1, L9). This distribution coverage metric will be used for model tuning and checkpoint selection (Alg 1, L16). At each pseudo-labeling step, given a pseudo-labeled triplet $(v_-, q_+, \arg\max(p_-))$ that satisfies the entropy metric, a labeled training triplet $(v_+, q_+, a_+)$ will also be sampled to be trained with equal weighting (Alg 1, L10). This weight balancing allows the model to learn from both image distributions, and acts as a model correctness regularization.

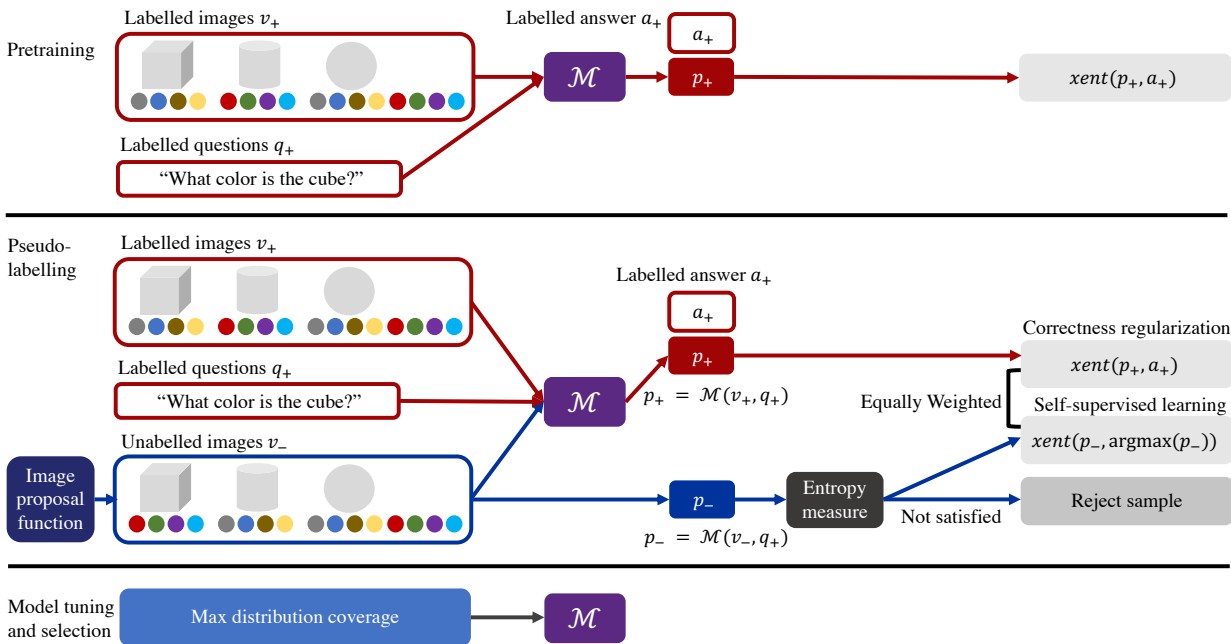

Figure 2: Overview of the DisCo framework. At each pseudo-labeling step, labeled VQA triplets from the train distribution are used in an equal weighting to pseudo-labeled VQA triplets, which contain images from the test distribution of novel attribute compositions.

### 3.3   Image proposals

DisCo is compatible with various kinds of unlabeled image distributions. In this paper, we focus on two prevalent choices for $f$: 1) direct sampling from unlabeled images in $D_{\text{test}}$, and 2) generation from a generative adversarial network (GAN) trained on unlabeled images in $D_{\text{test}}$.

We can directly sample unlabeled images from $D_{\text{test}}$ and propose each image as a potential pseudo-label target. We show in experiments that this method achieves strong performance on $D_{\text{test}}$, as well as on an unseen dataset that has the same distribution as $D_{\text{test}}$—both of which are *out-of-distribution* compared to the labeled set $D_{\text{train}}$. A potential approach to better cover test image distribution is to use generative models. We first train a GAN on unlabeled test images and make inferences of the trained model for image proposals, which essentially acts as a data augmentation. In this work, we train an unconditional StyleGAN v2 (Karras et al., 2020) on images from $D_{\text{test}}$. We also show that both sampling methods achieve improved performance on a setting with an *unknown* test distribution, where unlabeled images are drawn from a distribution that is different from both the training and test set.

### 3.4   Pseudo-label selection with entropy threshold

Pseudo-labeling out-of-distribution test examples in visual reasoning is especially noisy and challenging. This is not only because unlabeled images contain novel attribute compositions correlated with visual challenges such as occlusion, but also because randomly-sampled questions may contain presuppositions that the images must satisfy. That is, the referred objects in the question may not exist in the image. When the vocabulary of concepts is large, most randomly sampled images and question pairs will be unanswerable. Without an effective measure for filtering the pseudo-labeled training data, the model will be corrupted with a high percentage of inaccurate labels that gives a poor signal in generalization.

Recall that there are two types of errors we want to filter out. The first is presupposition failure, where the referred object in $q_+$ does not exist in $v_-$. For example, the question asks "what size is the cylinder", but

there is no cylinder in the image. The second is questions that are difficult to answer due to limited training data or other visual challenges. For example, questions regarding a novel color-shape combination of a gray cylinder may be difficult to answer given the partial obstruction of the object. Below, we introduce and validate an entropy-based measure, which effectively filters both types of errors.

In this work, we leverage a strong correlation between the entropy of $p \leftarrow \mathcal{M}(v_-, q_+; \theta)$, and compositional generalization accuracy. We demonstrate that this metric is an effective measure for pseudo-labeling out-of-distribution images from the test distribution with unseen attribute compositions. The entropy is calculated from the softmax of model logits. Let $k$ denote the number of elements in the output vocabulary, the entropy is computed as $\mathcal{H}(X) = -\sum_{i=1}^{k} p(x_i) \log p(x_i)$.

In Figure 3, we empirically verify this relationship between entropy and question-answering accuracy on images with novel attribute combinations. The left graph 3a) shows a cumulative entropy to accuracy plot on a log scale, with presupposition adherence accuracy, prediction accuracy, and prediction accuracy given presupposition adherence. Presupposition adherence accuracy (blue), is the percentage of question-image pairs whose unlabeled image satisfies the sampled question's presupposition (i.e., the referred-to object exists). Prediction accuracy (red) is calculated such that the presupposition is satisfied *and* the predicted answer is correct. Prediction accuracy given presupposition adherence (purple), is the prediction accuracy of only pairs that adhere to presupposition. The right graph 3b) depicts a cumulative entropy histogram.

Interestingly, we find that while prediction accuracy for images that satisfy presuppositions (purple) does not decrease significantly with entropy increase, the percentage of question-image pairs that violate presuppositions (blue) does decrease significantly. This suggests that our entropy measure well captures presupposition failure, and is hence effective and necessary for this learning paradigm. The black line on both graphs indicates an approximately 30-th percentile entropy threshold, based on the histogram of entropies, which yields a 0.8207 prediction accuracy on out-of-distribution test images and a 0.8506 presupposition adherence accuracy. Given images that passed the question presupposition at this threshold, 0.9648 were accurate.

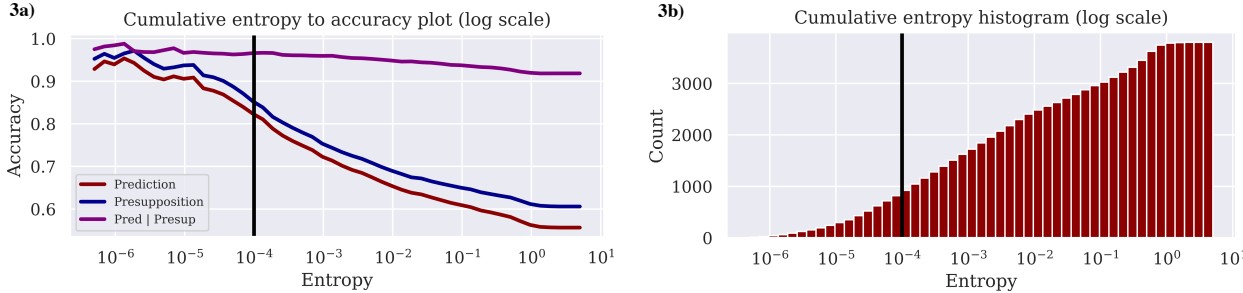

Figure 3: Relationship between entropy and compositional generalization accuracy and count. The base VQA model is trained on a biased train dataset; entropy and prediction accuracy is evaluated on unlabeled images sampled from the test dataset paired with questions from the train set.

## 3.5 Model selection

For compositional generalization tasks, because we do not have access to the ground-truth labels for images from the test distribution, there is no natural criterion for model tuning and model selection. A common practice of previous methods for model selection is through the maximization of validation set accuracy; however, the validation set has the same data distribution as the train set, and thus is biased and not a good measure for test distributions. In this work, we propose a more effective measure by leveraging the unlabeled image set for model tuning and model selection.

Specifically, DisCo employs a distribution-coverage-based metric, which does *not* require any labeled examples from the test distribution. The high-level idea is to maximize the distribution coverage on the unlabeled dataset. Formally, recall that during training, the sampler produces a pair $(v_-, q_+)$ for pseudo-labeling,

where $v_-$ is from the unlabeled distribution and $q_+$ is from the train set. Our algorithm keeps track of the percentage of pairs $(v_-, q_+)$ that are rejected from the entropy thresholding for each model (checkpoint). After training, we select the model with the maximum coverage of the test set, i.e., the model that rejects the least number of pseudo-labeling pairs $(v_-, q_+)$.

Empirically, we validate that distribution coverage from our framework well approximates compositional generalization accuracy in the test set, while the standard validation set accuracy does not. In Figure 4, the left plot 4a) depicts the correlation between test set accuracy and validation set accuracy from model checkpoints, while the right plot 4b) depicts the correlation with distribution coverage. The Pearson correlation coefficient between validation and test set accuracy is 0.2565. In this experiment, we can see that validation set accuracies are mostly close to 1.0, while test set accuracies range from 0.90 to 0.98. In comparison, the distribution coverage value has a strong correlation with the test performance: the Pearson correlation coefficient is 0.6066, and thus is a more effective metric for model selection.

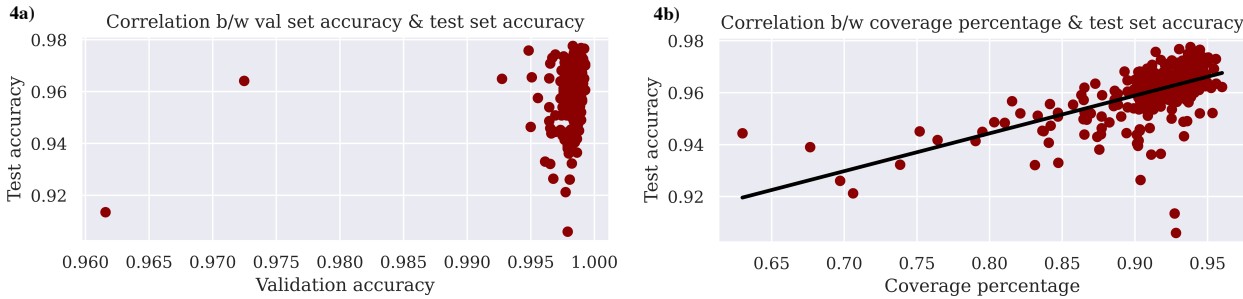

Figure 4: Correlation between test set and validation set accuracy (left plot, with Pearson correlation coefficient of 0.2565) as well as between test set accuracy and our distribution coverage metric (right plot, with a coefficient of 0.6066).

## 4 Experiments

We evaluate DisCo on a set of CLEVR datasets and three visual question-answering models—FiLM (Perez et al., 2018), a representative end-to-end attention-based approach, TbD-net (Mascharka et al., 2018), a state-of-the-art neural module network-based approach, and NS-CL, a neuro-symbolic and object-centric approach. DisCo considerably improves the compositional generalization performance of base models compared to their original training paradigm or other semi-supervised learning approaches. Specifically, we compare our framework against two baselines that leverage unlabeled data: variational autoencoders (VAEs; Kingma & Welling, 2014) and SimCLR (Chen et al., 2020). For both baselines, we first train a VAE or a SimCLR model, and use their encoding networks to initialize the feature extractor of the visual reasoning model. Both baselines use the same amount and exact set of unlabeled data as DisCo. We describe our datasets and implementation details in Section 4.1, compare our work against prior work in Section 4.2, and provide more ablation studies in Section 4.3, and more analyses in Section 4.4.

### 4.1 Datasets & implementation details

In addition to the original CLEVR compositional generalization (CoGenT) dataset [1] (Johnson et al., 2017) (released under the CC BY 4.0 license), we also report results on multiple CoGen datasets based on CLEVR. Specifically, we generate images with two to three objects per image, with a CoGen split following that of the original CLEVR dataset. In this setup, there are two sets of colors, with the first as [*gray, blue, brown, yellow*], and the second as [*red, green, purple, cyan*]. CoGen split $A$ contains cubes in the first set of colors,

---

[1]We use the term CoGenT to specifically refer to the original compositional generalization dataset introduced in Johnson et al. (2017), and use the term CoGen to generally refer to dataset splits (e.g., CoGen split $A$, CoGen split $B$) generated for testing compositional generalization.

Table 1: DisCo performance compared to the original training paradigm and baselines, where 0.5% Ref (unk) is performance on an unknown test distribution, with the unlabeled dataset not drawn from the specified test distribution, and 0.5% Ref (unseen) is performance on an unseen test set not exposed during training .

|  | 0.0% REF | 0.1% REF | 0.5% REF | 0.5% REF (UNK) | 0.5% REF (UNSEEN) | 0.5% ONEHOP | 0.5% COGENT |
|---|---|---|---|---|---|---|---|
| FILM | 0.7589 | 0.7931 | 0.9265 | 0.9265 | 0.9251 | 0.9270 | 0.7859 |
| FILM + VAE | 0.7500 | 0.7993 | 0.9387 | 0.9396 | 0.9362 | 0.9228 | 0.7868 |
| FILM + SIMCLR | 0.7520 | 0.8036 | 0.9288 | 0.9255 | 0.9257 | 0.9261 | 0.7926 |
| **FiLM + DisCo-S** | **0.7582** | **0.8363** | **0.9621** | **0.9609** | **0.9616** | **0.9469** | **0.8004** |
| **FiLM + DisCo-G** | **0.7760** | **0.8191** | **0.9545** | **0.9524** | **0.9510** | **0.9311** | **0.7979** |

Table 2: Comparison of DisCo with baselines on the TbD-net model. Performance reported on the original, unknown, and unseen test set of 0.5% Ref.

|  | 0.5% REF | 0.5% REF (UNK) | 0.5% REF (UNSEEN) |
|---|---|---|---|
| TBD | 0.8993 | 0.8993 | 0.9027 |
| TBD + VAE | 0.9018 | 0.9027 | 0.9025 |
| TBD + SIMCLR | 0.9073 | 0.9054 | 0.9077 |
| **TbD + DisCo-S** | **0.9206** | **0.9205** | **0.9197** |
| **TbD + DisCo-G** | **0.9189** | **0.9156** | **0.9145** |

Table 3: Comparison of DisCo with baselines on the NS-CL model. Performance reported on the original, unknown, and unseen test set of 0.5% Ref.

|  | 0.5% REF | 0.5% REF (UNK) | 0.5% REF (UNSEEN) |
|---|---|---|---|
| NS-CL | 0.7622 | 0.7622 | 0.7633 |
| NS-CL + VAE | 0.7572 | 0.7583 | 0.7589 |
| NS-CL + SIMCLR | 0.7739 | 0.7727 | 0.7758 |
| **NS-CL + DisCo-S** | **0.8024** | **0.8016** | **0.8027** |
| **NS-CL + DisCo-G** | **0.7820** | **0.7798** | **0.7793** |

and cylinders in the second. CoGen split $B$ is reversed in the attribute combinations. Both CoGen split $A$ and $B$ contain spheres of all eight colors. We study a biased setup, where $D_{\text{train}}$ consists of images in CoGen split $A$ with either zero or only a small percentage $p$ of objects from CoGen split $B$. We show performance with train datasets of $p \in \{0.0, 0.001, 0.005\}$, evaluated on a full CoGen split $B$ test set.

Based on the aforementioned image setup, we generate two additional datasets with different types of questions. The first is the referred object dataset (*Ref*), with questions of the form, "What [attribute] is the [referred object]?" (e.g., "What material is the red object?"). Although this question template is simple, it reflects one of the most important problems in pseudo-labeling for VQA: the satisfaction of question presuppositions. The second dataset consists of "one-hop"[2] questions from the CLEVR dataset (*OneHop*), which consists of more complex relational questions, such as "How many red objects are to the left of the sphere?" or "There is a large object to the right of the metal thing; what is its color?". We also evaluate the models on a biased CLEVR CoGenT dataset (*CoGenT*) with the full set of complex objects and questions.

We use the official implementations of FiLM, TbD-net, and NS-CL along with their original hyperparameters in our framework. The GAN image proposal function is the unconditional StyleGAN2 (Karras et al., 2020), trained with the Adam optimizer of learning rate 0.002. We set our entropy threshold $n$ to be at the 30th percentile. Empirically, we find this threshold value to be robust to differences of around 10 percentile increase or decrease. For the more complex CoGenT dataset, we lowered the entropy threshold to be at the 10th percentile to account for the naturally lower percentage of presupposition adherence. All models are trained on a single Titan RTX GPU.

## 4.2 Results

We train DisCo with FiLM on five datasets—three Ref datasets with varying bias levels, from 0.0% (fully biased), to 0.1% biased, to 0.5% biased, as well as the 0.5% biased OneHop dataset and 0.5% biased CoGenT

---

[2]In *one-hop* questions, the target objects are referred to by relating to another unique object.

dataset. We show the results of our framework with both image proposal functions (DisCo-S as direct sampling and DisCo-G as GAN generation). Note that for VAE, SimCLR, and our framework DisCo, the VQA models see unlabeled test set images during pretraining or pseudo-labeling. Thus, for a fair comparison, we additionally report accuracy on a larger, test set that contains unseen images following the same distribution as the original test set, for 0.5% biased Ref. In addition, while we mainly focus on experiments that leverage unlabeled data from a known test distribution to improve the performance of that specific distribution shift, we also present results in a setting where the test dataset is from an unknown test distribution. In this setting, the unlabeled data is drawn from an expanded set of distributions including the test distribution, but not solely consisting of samples that follow the test distribution. We set up this experiment with the training dataset from CoGen split *A*, the test dataset from CoGen split *B*, and the unlabeled dataset from the full CLEVR split consisting of both split *A* and *B* with all color-shape attributes.

Table 1 shows our results; DisCo outperforms the original training paradigm and both baselines. In addition, our framework, with both image proposal functions, is robust to both images from an unknown test distribution as well as unseen images in a known test distribution. By comparing the two image proposal approaches across experiments, DisCo-S achieves better performance than DisCo-G. We conjecture that this is because, in the sampler method, the model is trained with the exact set of *real*, unlabeled test images that we evaluate with. Moreover, in the fully biased Ref experiments, the GAN image proposal function outperforms the direct sampler. We attribute this to the GAN's generation of more diverse unlabeled images, which can better cover the image space of possible camera angles and lighting conditions, allowing DisCo to improve model performance when there are few signals from the labeled VQA dataset. The fully biased Ref experiments illustrate a difficult and important setting; there are often cases where it is useful to perform well without any labeled data in the compositional generalization setting, e.g. when we only have access to objects with a specified set of attributes during training and do not know the test distribution, or when we need to adapt to new test distributions without labels.

Table 4: Ablation of DisCo with direct sampler, trained on the 0.5% bias Ref. (PL = pseudo-labeling, EM = entropy measure, CS = coverage selection).

|  | FILM |
|---|---|
| BASE | 0.9265 |
| BASE+PL | 0.6749 |
| BASE+PL+EM | 0.9605 |
| BASE+PL+EM+CS | **0.9621** |

**Integration with TbD-net and NS-CL.** DisCo can be integrated with other visual reasoning models, too. In this paper, we implement TbD-net and NS-CL with DisCo, to showcase flexibility. The results on TbD-net are presented on the 0.5% Ref split in Table 2, and results on NS-CL in Table 3. The observations are consistent with the FiLM-based experiments. DisCo similarly outperforms the original training paradigm and baselines by a noticeable margin.

### 4.3 Ablations

In Table 4, we present ablation studies of our framework with a direct sampler as the image proposal function. First, we see that directly adding the pseudo-labeling (PL) module (i.e., pseudo-label all images in the test set with the pretrained FiLM model without any thresholding) significantly degrades model accuracy. Second, adding our entropy measure (EM) improves compositional generalization performance. This finding is consistent with our visualizations of the correlation between entropy and accuracy in Fig. 3. Moreover, adding coverage-based selection (CS) further improves test accuracy.

Table 5 zooms in into the candidate model selection methods. Specifically, we compare test accuracy from our coverage-based selection strategy (Coverage) with test accuracy from the standard, validation accuracy-based selection strategy (Val acc). Our framework shows consistent advantage across FiLM, TbD-net, and NS-CL VQA models. Note that the distribution coverage metric is not directly applicable to pretraining-based baselines (VAE and SimCLR) because they do not compute pseudo-labels for test images.

### 4.4 Analyses

**Qualitative examples** We qualitatively analyze the performance gain brought by our framework. Figure 5 (top row) shows two example images in the CoGen split *B* test set. We apply the base FiLM model, the

Table 5: Comparison of test accuracy with baselines on model selection strategy, trained on 0.5% Ref.

|  | FiLM | | TbD-net | | NS-CL | |
|---|---|---|---|---|---|---|
|  | Val acc | Coverage | Val acc | Coverage | Val acc | Coverage |
| Base | 0.9265 | N/A | 0.8993 | N/A | 0.7622 | N/A |
| Base + vae | 0.9387 | N/A | 0.9018 | N/A | 0.7572 | N/A |
| Base + simclr | 0.9288 | N/A | 0.9073 | N/A | 0.7739 | N/A |
| **Base + DisCo-S** | 0.9605 | **0.9621** | 0.9000 | **0.9206** | 0.7887 | **0.8024** |
| **Base + DisCo-G** | 0.9508 | **0.9545** | 0.9020 | **0.9189** | 0.7780 | **0.7820** |

FiLM model trained with DisCo-S (direct sampling), and the FiLM model trained with DisCo-G (GAN) on the fully biased Ref dataset and retrieve their predictions.

In the top left example, the test set question asks "What color is the cylinder?" of the *brown cylinder* in the image. Brown cylinders are never seen in the fully biased train set, hence the FiLM model answers incorrectly with *red*, unable to identify the referred object. DisCo with direct sampling also produces a wrong answer, likely due to a lack of signal from the labeled image set to bootstrap visual reasoning. In this case, we see that DisCo with GAN is able to answer correctly with *brown* and show better compositional generalization. We conjecture that this is due to the GAN covering a denser image distribution. In the bottom row of Figure 5, we see two similar images— the left image taken from the train set with a *brown cube* and cyan cylinder, and the right image generated by our GAN with a *brown cylinder* in a closely aligned scene. We hypothesize that it is diverse image proposals like this that enable visual reasoning models to better learn the concept of a *brown cylinder.*

In the top right example of Figure 5, similarly, *purple cubes* are never seen in the fully biased train set, thus the FiLM model answers a completely incorrect color, while DisCo with both image proposal functions is able to generalize to this novel attribute combination.

**Limitations** DisCo provides a framework for bootstrapping visual reasoning; however, it relies on the inductive bias of convolutional networks to compositionally generalize. Without priors on visual attributes, given a *red cube* (a novel color-shape combination not seen in the train set), the model could learn that the object is neither *red* nor a *cube*—as *red* could be learned as a color only existing on cylinders and spheres, while *cube* could be learned as a shape that is only paired with colors that are gray, blue, brown, or yellow, as seen from the labeled train set. Assumptions about inductive biases of visual attributes in convolutional networks allow our framework to learn generalization, and hence DisCo is limited to, and also especially powerful, in the vision domain.

We do not report test results on more real-world datasets due to the lack of datasets that evaluate compositional generalization. This leaves several open questions for future work, such as how DisCo may perform with more variability in the dataset, and with a base model that exhibits poor generalization ability and is challenging to bootstrap from. Additionally, the effects of utilizing a thresholding metric based on variants of the softmax score (Hendrycks & Gimpel, 2017; Liang et al., 2018) and the energy score (Liu et al., 2020) may also be fruitful to explore as future work.

What color is the cylinder? What color is the cube?

Ground truth: brown
FiLM: red
FiLM + DisCo-S: red
FiLM + DisCo-G: brown

Ground truth: purple
FiLM: blue
FiLM + DisCo-S: purple
FiLM + DisCo-G: purple

Image from train set  GAN image proposal

Figure 5: Top row: two prediction examples from FiLM, DisCo-S, and DisCo-G on fully biased Ref. Bottom row: two closely aligned images from the train set and from the GAN proposal function.

# 5 Conclusion

We have presented DisCo, a framework for improving compositional generalization by leveraging *unlabeled, out-of-distribution* images from the test distribution through iterative pseudo-labeling. We studied and proposed the entropy measure as an effective signal for presupposition adherence and pseudo-label accuracy in out-of-distribution test examples, and also introduced the distribution coverage model selection strategy, which well approximates test performance on novel attribute combinations while only requiring unlabeled data. We demonstrated our framework's ability to improve compositional generalization performance, and showed potential for future work to leverage unlabeled images to achieve generalization in evaluation regimes with combinatorial complexity.

## Broader Impact Statement

Our work shows the importance of learning unbiased concepts from datasets with better distribution coverage. We expect minimal negative societal impact, however, when using our framework, it's important to ensure that the unlabeled dataset itself has enough distribution coverage to minimize dataset bias. Our goal is for DisCo to help models perform well in data-limited environments for good.

## Acknowledgments

We thank Eric Chan for providing valuable feedback on the paper. This work is in part supported by the Stanford Institute for Human-Centered AI (HAI), Toyota Research Institute (TRI), NSF RI #2211258, ONR MURI N00014-22-1-2740, and Analog, JPMC, Meta, Salesforce, and Samsung. JH is supported by the Knight Hennessy fellowship and the NSF Graduate Research Fellowship.

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

# A  Appendix

The supplementary material is organized as the following. First, in Section A.1, we provide released code, and in Section A.2, we describe our dataset construction. Section A.3 shows results from our image proposal functions, while Section A.4 and Section A.5 demonstrate ablations on entropy thresholds and robustness of the distribution coverage metric. We show additional experiment results on settings with an unknown test distribution in Section A.6. In Section A.7, we provide details on baseline implementations. Last, in Section A.8 and Section A.9, we report quantitative analyses of our model performance as well as qualitative examples.

## A.1  Code release

Code for DisCo with the FiLM model can be found: https://github.com/joyhsu0504/disco, based on the FiLM codebase (https://github.com/ethanjperez/film). We want to highlight that when using DisCo, it is important to ensure that the unlabeled dataset has enough distribution coverage to minimize dataset bias.

## A.2  Dataset

We generate additional CLEVR datasets of two to three objects based on the CoGen (compositional generalization) split introduced in Johnson et al. (2017). As a recap, in this setup, there are two sets of colors, with the first as [*gray, blue, brown, yellow*], and the second as [*red, green, purple, cyan*]. CoGen split *A* contains cubes in the first set of colors, and cylinders in the second. CoGen split *B* is reversed in the attribute combinations. Both CoGen split *A* and *B* contain spheres of all eight colors. In our construction, the train set of CoGen split *A* consists of 8,000 images, and the validation set of CoGen split *A* and the test set of CoGen split *B* consist of 2,000 images each. The larger, unseen test set consists of 8,000 images. The Ref datasets include questions of the form "What [attribute] is the [referred object]?", while the OneHop dataset includes one-hop relation questions as defined in Johnson et al. (2017).

### A.3 Image proposal examples

In Figure 6, we provide examples of image proposals from our two functions—direct sampling and GAN generation. Both capture a range of novel attribute combinations not in the labeled train set images.

Direct sampling image proposals                GAN generated image proposals

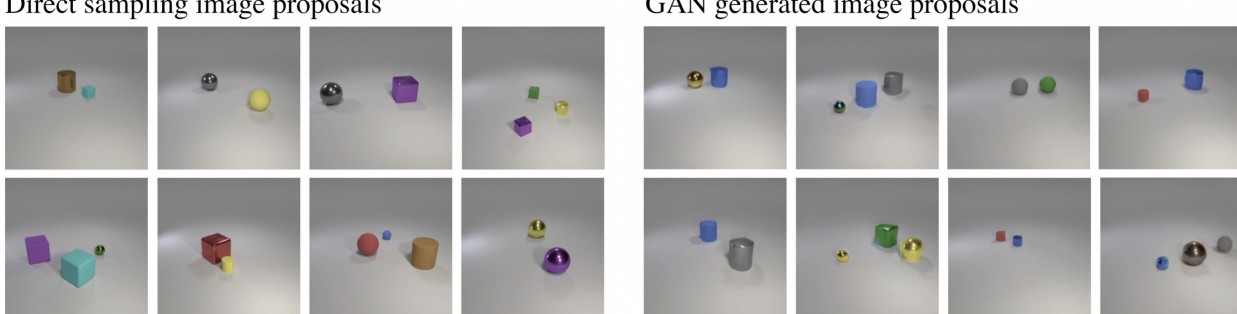

Figure 6: Image proposals from direct sampling and GAN generation, used as unlabeled image input in DisCo.

### A.4 Entropy thresholds

In this ablation study, we validate the robustness of DisCo with respect to different entropy thresholds. We show that DisCo yields strong results at different values of the hyperparameter. See Table 6 for results of FiLM with DisCo-S on entropy thresholds at varying percentiles. In addition, we present test accuracy curves for each run, compared to the base FiLM model in Figure 7. The black line indicates where DisCo begins pseudo-labeling from the pretrained FiLM model. We observe that DisCo considerably improves upon the base VQA model. These results also show the robustness of our method against different random seeds. For each run shown, the network weights, as well as the data samples, are generated based on different random seeds, but the improvements are consistent.

Table 6: Comparison of different percentiles of entropy thresholds on the 0.5% biased Ref dataset.

|  | 30TH | 35TH | 40TH | 45TH |
|---|---|---|---|---|
| FiLM + DisCo-S | 0.9612 | 0.9616 | 0.9667 | 0.9643 |

### A.5 Distribution coverage

We demonstrate that our distribution coverage measure well approximates test accuracy, even when the VQA model is decreasing in performance. In Figure 8, we report the test accuracy and distribution coverage curves of a FiLM + DisCo-S experiment at too high an entropy threshold, where performance quickly degrades. In Figure 9, we present the curves of a FiLM + DisCo-G experiment that slightly degrades in performance before recovering. The Pearson correlation coefficient between test accuracy and distribution coverage for these experiments are 0.6585 and 0.7987, respectively, both showing highly correlated values allowing for effective model selection.

We see that DisCo does not tend to oversample or propagate errors; instead, it lowers the number of samples chosen when compositional generalization performance decreases. We conjecture that this is due to our model correctness regularization of training on labeled triplets, which contain different combinations of attributes. When the base VQA model is trained on corrupted pseudo-labels, the model is no longer confident in its predictions given conflicting signals from labeled and pseudo-labeled triplets, and hence fewer samples are chosen, as intended in our framework.

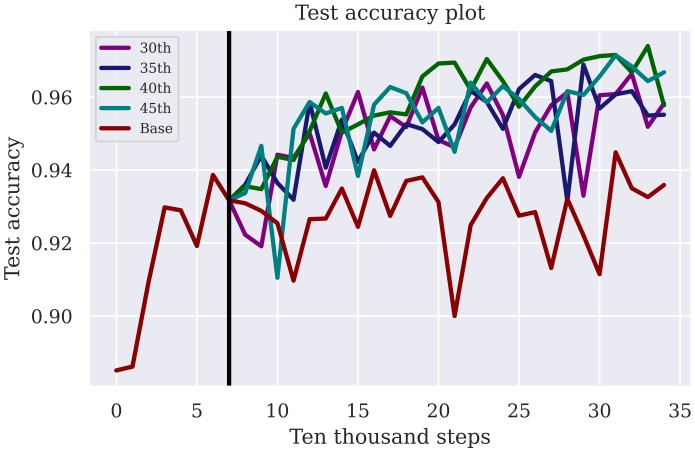

Figure 7: Test accuracy curves of the base FiLM model (red), and DisCo-S at different percentiles of entropy thresholds.

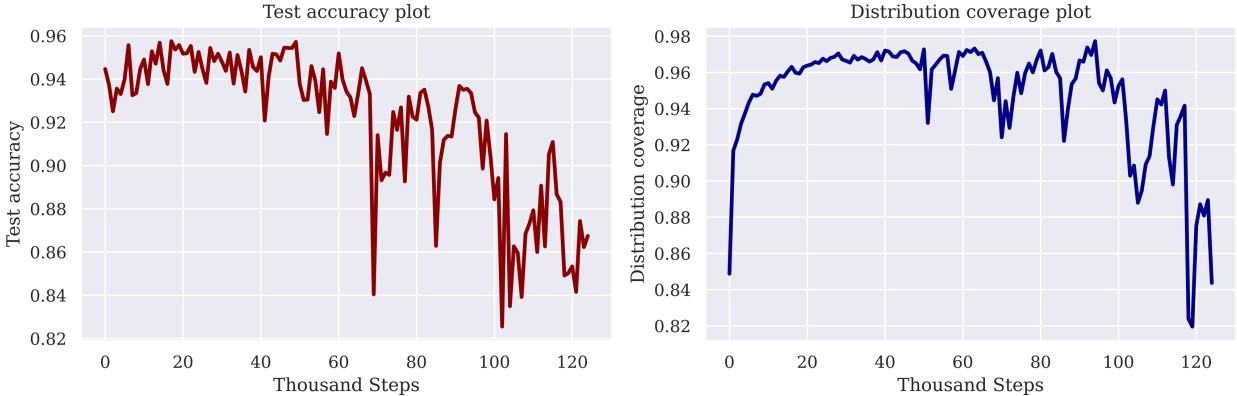

Figure 8: Test accuracy and distribution coverage curves of an experiment with large decrease in model performance, with Pearson correlation coefficient of 0.6585.

### A.6 Experiments with unknown test distributions

Although we mainly focus on settings where the unlabeled dataset is drawn from a known test distribution, we also show experiments on settings where the test distribution is unknown. We report results on settings where the unlabeled dataset is a superset of the unknown test distribution (UNK in the main text), as well as on settings where the unlabeled dataset is a subset of the unknown test distribution (UNK-SUB below). In the latter setup, we train DisCo on labeled CoGen split $A$, leverage unlabeled CoGen split $B$, and test model performance on the full CLEVR split. We see in Table 7, Table 8, and Table 9 that DisCo shows consistent performance gain compared to base and baseline models in this setting for all three VQA backbones.

### A.7 Baseline implementation

We implemented two pretraining approaches that leverage unlabeled images. Specifically, we first train a variational autoencoder (VAE; Kingma & Welling, 2014) and SimCLR (Chen et al., 2020) model and use their encoding networks to initialize the feature extractor of the visual reasoning model. The VAE implementation is based on https://github.com/AntixK/PyTorch-VAE, and the SimCLR implementation on https://github.com/Spijkervet/SimCLR. For both methods, we use retrieved image features from ResNet101

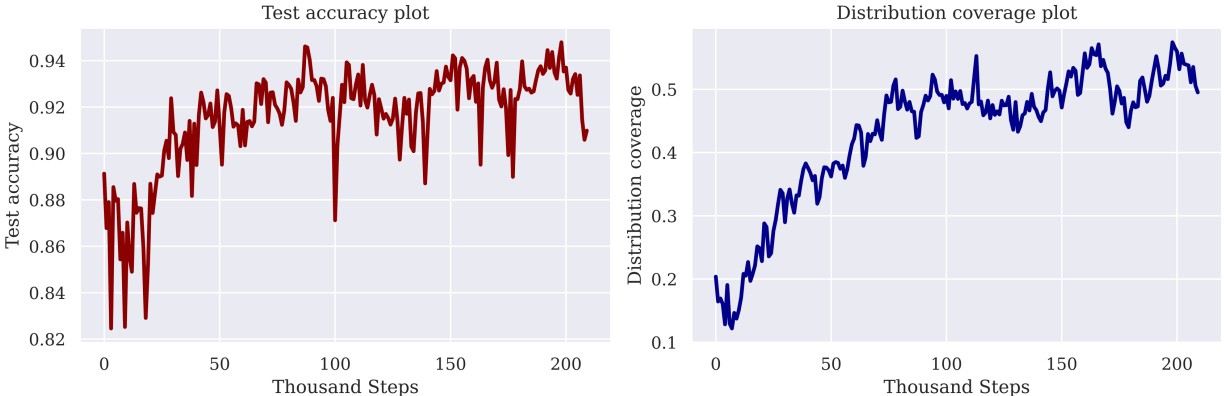

Figure 9: Test accuracy and distribution coverage curves of an experiment with a slight decrease in model performance, with Pearson correlation coefficient of 0.7987.

Table 7: DisCo on the FiLM model. Performance reported on a test set of 0.5% Ref from an unknown test distribution.

Table 8: DisCo on the TbD-net model. Performance reported on a test set of 0.5% Ref from an unknown test distribution.

Table 9: DisCo on the NS-CL model. Performance reported on a test set of 0.5% Ref from an unknown test distribution.

|  | 0.5% REF (UNK-SUB) |
|---|---|
| FiLM | 0.9634 |
| FiLM + VAE | 0.9683 |
| FiLM + SIMCLR | 0.9635 |
| **FiLM + DisCo-S** | **0.9798** |
| **FiLM + DisCo-G** | **0.9745** |

|  | 0.5% REF (UNK-SUB) |
|---|---|
| TBD | 0.9506 |
| TBD + VAE | 0.9495 |
| TBD + SIMCLR | 0.9510 |
| **TBD + DisCo-S** | **0.9582** |
| **TBD + DisCo-G** | **0.9536** |

|  | 0.5% REF (UNK-SUB) |
|---|---|
| NS-CL | 0.8810 |
| NS-CL + VAE | 0.8789 |
| NS-CL + SIMCLR | 0.8872 |
| **NS-CL + DisCo-S** | **0.9011** |
| **NS-CL + DisCo-G** | **0.8902** |

as input to the VAE and SimCLR model. We add in encoding layers of [Conv2d, BatchNorm2d, and LeakyReLU] to both networks, and use feature-level reconstruction and contrastive loss to supervise learning. After pretraining, we use the newly added layers as the additional encoding for our VQA models.

## A.8 Quantitative analyses

In Table 10, we examine the test set accuracy per color-shape combination of referred objects in the fully biased Ref dataset. We compare FiLM with DisCo-S and DisCo-G, and report metrics on attribute combinations not seen in the labeled train set.

Interestingly, the best-performing model with GAN-generated image proposals performs significantly better on cubes, with a decrease in accuracy on some cylinders in comparison to FiLM and direct sampling. We might instead expect uniformly increased performance on all color-shape combinations, but empirical results reveal that models that compositionally generalize may learn to do so better on some set of novel attribute combinations.

## A.9 Qualitative examples

In Figure 10, we show examples of VQA pairs at different values of entropy output by the model. We see that low entropy examples both satisfy presuppositions and correctness, while at a higher entropy value,

Table 10: Comparison of test accuracy per color-shape combination of the referred object, trained on the fully biased Ref dataset.

| | FILM | FILM + DISCO-S | FILM + DISCO-G |
|---|---|---|---|
| RED CUBE | 0.6794 | 0.6589 | 0.7658 |
| GREEN CUBE | 0.6399 | 0.638 | 0.7173 |
| PURPLE CUBE | 0.6478 | 0.6478 | 0.7676 |
| CYAN CUBE | 0.6313 | 0.6313 | 0.6774 |
| GRAY CYLINDER | 0.6481 | 0.6490 | 0.6292 |
| BLUE CYLINDER | 0.6431 | 0.6450 | 0.5752 |
| BROWN CYLINDER | 0.6654 | 0.6774 | 0.6719 |
| YELLOW CYLINDER | 0.6385 | 0.6459 | 0.6106 |
| RED SPHERE | 1.0 | 0.994 | 0.9859 |
| GREEN SPHERE | 1.0 | 0.9937 | 0.9958 |
| PURPLE SPHERE | 0.9958 | 0.9958 | 0.9917 |
| CYAN SPHERE | 1.0 | 1.0 | 0.9981 |
| GRAY SPHERE | 0.9948 | 0.9923 | 0.9794 |
| BLUE SPHERE | 1.0 | 1.0 | 0.9961 |
| BROWN SPHERE | 1.0 | 1.0 | 0.9981 |
| YELLOW SPHERE | 1.0 | 0.9980 | 0.9940 |

only presuppositions are satisfied but the answer predicted by the model for out-of-distribution objects is incorrect, and at the highest values of entropy there exists presupposition failure. In this way, DisCo is able to choose suitable pairs to be added to training.

We also present additional qualitative examples of predictions from FiLM, DisCo-S, and DisCo-G. In Figure 11, we see examples where DisCo-S and DisCo-G outperform FiLM (first row) and examples where DisCo-G outperforms DisCo-S and FiLM (second row), on the fully biased Ref dataset.

| What size is the cyan thing? | What shape is the rubber thing? | What shape is the small thing? | What material is the cylinder? | What color is the sphere? |
|---|---|---|---|---|

Entropy: 0.0000000009
Presupposition: ✔
Correct: ✔
Ground truth: small
DisCo-S: small

Entropy: 0.0000042114
Presupposition: ✔
Correct: ✔
Ground truth: cube
DisCo-S: cube

Entropy: 0.0001019438
Presupposition: ✔
Correct: ✘
Ground truth: cylinder
DisCo-S: cube

Entropy: 0.0019916531
Presupposition: ✔
Correct: ✘
Ground truth: metal
DisCo-S: rubber

Entropy: 0.6848732829
Presupposition: ✘
Correct: ✘
Ground truth: DNE
DisCo-S: green

Figure 10: Qualitative examples VQA pairs at different levels of entropy.

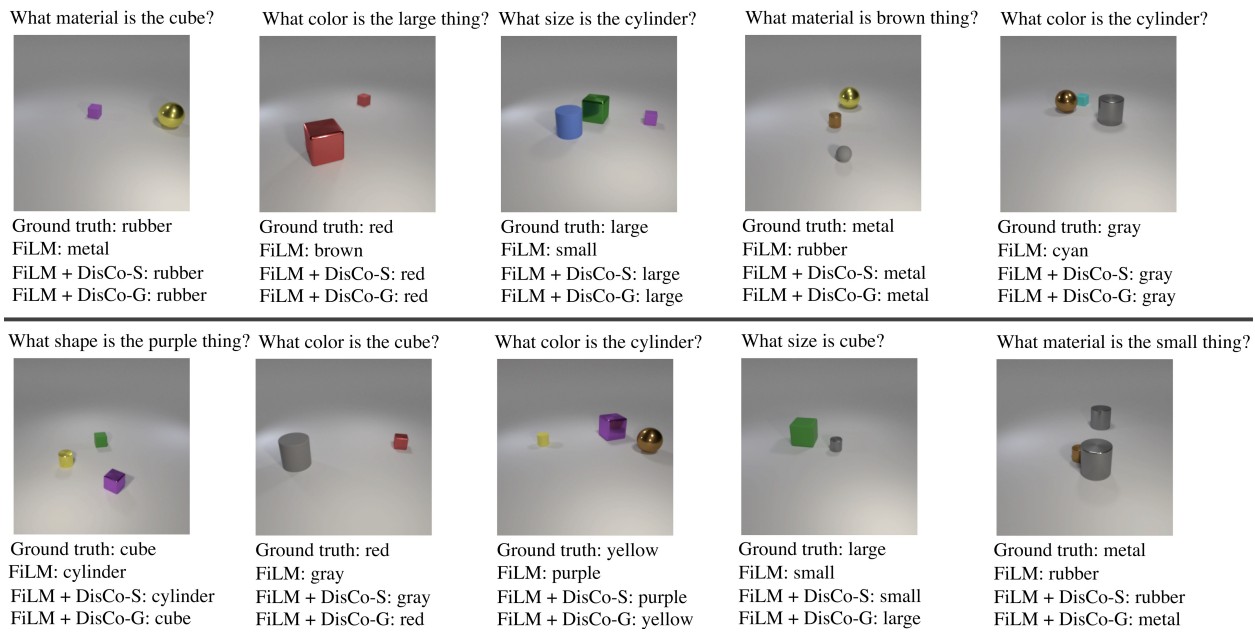

What material is the cube?
Ground truth: rubber
FiLM: metal
FiLM + DisCo-S: rubber
FiLM + DisCo-G: rubber

What color is the large thing?
Ground truth: red
FiLM: brown
FiLM + DisCo-S: red
FiLM + DisCo-G: red

What size is the cylinder?
Ground truth: large
FiLM: small
FiLM + DisCo-S: large
FiLM + DisCo-G: large

What material is brown thing?
Ground truth: metal
FiLM: rubber
FiLM + DisCo-S: metal
FiLM + DisCo-G: metal

What color is the cylinder?
Ground truth: gray
FiLM: cyan
FiLM + DisCo-S: gray
FiLM + DisCo-G: gray

What shape is the purple thing?
Ground truth: cube
FiLM: cylinder
FiLM + DisCo-S: cylinder
FiLM + DisCo-G: cube

What color is the cube?
Ground truth: red
FiLM: gray
FiLM + DisCo-S: gray
FiLM + DisCo-G: red

What color is the cylinder?
Ground truth: yellow
FiLM: purple
FiLM + DisCo-S: purple
FiLM + DisCo-G: yellow

What size is cube?
Ground truth: large
FiLM: small
FiLM + DisCo-S: small
FiLM + DisCo-G: large

What material is the small thing?
Ground truth: metal
FiLM: rubber
FiLM + DisCo-S: rubber
FiLM + DisCo-G: metal

Figure 11: Qualitative examples of FiLM, DisCo-S, and DisCo-G on the fully biased Ref dataset.

