# OpenReview forum: "DisCo: Improving Compositional Generalization in Visual Reasoning through Distribution Coverage"
_TMLR — Accepted by TMLR_

### Review · Reviewer_SCqd · 2022-11-29

**Summary Of Contributions:**

The paper proposes a learning method called: DisCo (Distribution Coverage) to improve compositional generalization for the CLEVR dataset. It proposes two components: the pseudo-labeling framework pairs up compositionally novel, unlabeled images with randomly selected questions to improve compositional generalization and the model selection strategy uses a distribution coverage metric to select models that generalize better. The results show improved compositional generalization in the CLEVR dataset.

**Audience:**

Yes

**Broader Impact Concerns:**

The existing section looks good.

**Claims And Evidence:**

Yes

**Requested Changes:**

It would be ideal to test the method on more realistic scenarios i.e., with varying types/degrees of shifts between the unlabeled and the actual test distributions preferably on datasets more realistic than CLEVR. I think such experiments could provide evidence for some of the claims with broad scope e.g., those made in Sec 3:

> As training progresses, DisCo selects more “difficult” question-image pairs, which increases distribution coverage on the unlabeled image set. After training, models can reason about objects with new attribute compositions, not seen with labels during training.

This is however not critical for acceptance as long as the claims are edited to reflect the scope/assumptions.

**Strengths And Weaknesses:**

**Strengths:**

[S1] Overall, the paper is clearly written; the proposed method is sensible and the results support the core claim of compositional generalization (if the scope of the claim is narrowed down (see weaknesses section)).

[S2] Both sampling and using a generative model are sound approaches to gathering unlabeled, test-like data.

[S3] Use of entropy threshold to filter out samples e.g., unanswerable ones is well motivated and Fig 3 does a really great job of justifying the use of entropy. This is also a simple measure to compute for practical purposes.

[S4] Model selection procedure is also sound as long as one has access to unlabeled test/test-like data. Sec 3.5 clearly shows the need and utility of this approach over a labeled val set that does not reflect test-like shifts.

**Weaknesses:**

The main weakness is that the experimental setup does not fully support the current broad claim of compositional generalization. To support the scope of the current claims, I think the paper needs to study different types and levels of shifts between unlabeled and test distributions, which is perhaps best studied with a more realistic dataset e.g., GQA instead of CLEVR. The other option would be to revise certain claims/sections.

---

> ### Author Response · Authors · 2022-12-13
> **Response to Reviewer SCqd**
>
> Thank you for the constructive feedback! Your review has helped us extend and refine the paper. We provide responses below.
>
> --------
>
> Request #1: Evaluation on more realistic datasets and different level of distribution shift between unlabeled data and test data.
>
> Response: Thank you for bringing up this point, which we fully agree. Based on your suggestion, we have included additional experiments on a different level of shift between the unlabeled dataset and test set distribution. In these experiments, we test on data from an unknown test distribution, where the unlabeled data is from an expanded set of distribution including the test distribution, but not solely consisting of samples that follow the test set distribution, as suggested. In this setup, the train dataset is from Clevr CoGen split A, the test dataset is from Clevr CoGen split B, and the unlabeled dataset is from the full Clevr split consisting of both split A and B with all color-shape attributes. We report results on this new setting (UNK experiments in Section 4.2) for all three base models — FiLM, TbD-net, and NS-CL. For each base model, we show comparisons against the two baselines pretrained on the new set of unlabeled data (VAE & Simclr) as well as our two DisCo sampling strategies derived from the new set of unlabeled data (DisCo-S and DisCo-G). The results are below:
>
> |                               | 0.5% Ref UNK    |
> | -----------                   | ----------- |
> | FiLM                          | 0.9265      |
> | FiLM + VAE                    | 0.9396      |
> | FiLM + SimCLR                 | 0.9255      |
> | FiLM + DisCo-S                | **0.9609**  |
> | FiLM + DisCo-G                | **0.9524**  |
>
> |                               | 0.5% Ref UNK    |
> | -----------                   | ----------- |
> | TbD-net                          | 0.8993      |
> | TbD-net + VAE                    | 0.9027      |
> | TbD-net + SimCLR                 | 0.9054      |
> | TbD-net + DisCo-S                | **0.9205**  |
> | TbD-net + DisCo-G                | **0.9156**  |
>
> |                               | 0.5% Ref UNK    |
> | -----------                   | ----------- |
> | NS-CL                          | 0.7622      |
> | NS-CL + VAE                    | 0.7583      |
> | NS-CL + SimCLR                 | 0.7727      |
> | NS-CL + DisCo-S                | **0.8016**  |
> | NS-CL + DisCo-G                | **0.7798**  |
>
> We see that DisCo reliably outperforms baselines in a setting with unknown test distribution. We have updated the main text with these descriptions and experiments in Section 4.2, as well as adjusted our claims to focus on the setting where we leverage unlabeled data from the test distribution to improve performance of that specific distribution shift.

---

> > ### Author Response · Authors · 2022-12-13
> > **Response to Reviewer SCqd (continued)**
> >
> > We also agree that it would be ideal to test DisCo on more realistic datasets along with CLEVR. To dive a bit deeper into our reasoning: We chose the CLEVR setup as our main evaluation setting, as it allows us to flexibly generate compositional generalization splits. Based on your suggestion, we looked into more realistic question answering datasets such as GQA-OOD, COVR, and VQA-CP [1, 2, 3]. We found that they are not currently suitable in our setup. GQA-OOD [1] tackles long-tailed distribution problems, while we study the compositional generalization setting. COVR [2] focuses on composition of reasoning steps in the language domain, instead of visual attributes. VQA-CP [3] contains train and test distributions that are approximately inverse, i.e., models become completely biased towards the train distribution while performance on the test distribution worsens, which forms a highly artificial relation [4]. DisCo implicitly assumes a setting where the base VQA model exhibits some generalization capability, such that DisCo can bootstrap and label more examples; in contrast, models trained on the training set of VQA-CP [3] are expected to fail on test distributions.
> >
> > That said, we indeed fully agree that testing on more realistic datasets is indeed a current limitation of our work, and have updated claims in the main text, as well as added a paragraph to our limitation section (Section 4.4) for discussion. In the added paragraph, we specify that we do not test on more realistic datasets in the paper due to lack of suitable datasets, and leave several open questions for future work.
> >
> > [1] Kervadec, C., Antipov, G., Baccouche, M., & Wolf, C. (2021). Roses are Red, Violets are Blue... But Should VQA Expect Them To?. In Proceedings of the IEEE/CVF Conference on Computer Vision and Pattern Recognition (pp. 2776-2785).
> >
> > [2] Bogin, B., Gupta, S., Gardner, M., & Berant, J. (2021). COVR: A Test-bed For Visually Grounded Compositional Generalization with Real Images. EMNLP’21.
> >
> > [3] Goyal, Y., Khot, T., Summers-Stay, D., Batra, D., & Parikh, D. (2017). Making the V in VQA Matter: Elevating the Role of Image Understanding in Visual Question Answering. In Proceedings of the IEEE conference on computer vision and pattern recognition (pp. 6904-6913).
> >
> > [4] Teney, D., Abbasnejad, E., Kafle, K., Shrestha, R., Kanan, C., & Van Den Hengel, A. (2020). On the Value of Out-of-distribution Testing: An Example of Goodhart's law. Advances in Neural Information Processing Systems, 33, 407-417.

---

> > > ### Comment · Reviewer_SCqd · 2022-12-24
> > > **Recommending Acceptance**
> > >
> > > Thank you for the response. The edits and the additional experiments address my concerns, so I am recommending acceptance.

---

> > > > ### Author Response · Authors · 2022-12-27
> > > > **Thanks to the reviewer**
> > > >
> > > > Thank you for your time and feedback on the paper!

---

### Review · Reviewer_h5vg · 2022-11-30

**Summary Of Contributions:**

The paper proposes to improve out of distribution performance by two main mechanisms. First, the use of pseudo-labeling and second, and more importantly, the use of unlabeled out of distribution data to improve distribution coverage. The experiments are conducted on the popolar CLEVR dataset with familiar backbone networks for CLEVR, and the results show improved compostional generalization to OOD samples in varying levels of distribution shifts introduced.

**Audience:**

Yes

**Claims And Evidence:**

Yes

**Requested Changes:**

- Both direct sampling and generative model should be sampled from an expanded set of distributions **INCLUDING** the test distribution, not solely consisting of samples that follow the test-set distribution. There should at least be a variant where the model should not have knowledge of the test-set distribution. The current experiments can be kept as-is but this should also be included. The use of samples generated from test distribution alone cast serious doubts on the claims made in the paper in the form that they appear currently. If the authors are unwilling to do this, the claims and writing should be changed to reflect the new setup (i.e., The model assumes direct and explicit knowledge of what exactly the "test" distribution looks like, OR, For known distribution shifts having unlabeled data can help improve performance on *that* specific distribution shift.). This is critical to securing my recommendation for accept.


**Strengths And Weaknesses:**

Strengths:

1. Technically sound and relevant to TMLR: I can find no obvious red flags with regard to technical correctness. The proposed entropy-based method for pseudo-label generation, or generation of samples using GAN, and the model selected using "distribution coverage" are all technically sound and can be expected to work in the manner proposed by the paper. The topic is also highly relevant to TMLR and should be of interest to a good subsection of its readers.

2. Thorough experiments: The experiments are clearly presented and are thorough enough to allow readers to know the effect of different choices used in the model. I do not see any holes or additional experiments that I would request. I do have a concern about whether the current experiments are enough to validate the main claims in the paper but they are a slightly orthogonal issue which I discuss below. This should not change the range of experiments presented, and I would still be totally okay with the same set of experiments repeated with a corrected training setup.

Weaknesses:

There is just one main weakness to me in the correct draft.

1. There are two main ways that the paper introduces unlabeled "OOD" samples to the proposed pipeline: A) direct sampling from unlabeled images in Dtest, and B) generation from a generative adversarial network (GAN) trained on unlabeled images in Dtest. Both of these are quite problematic and are antithetical to OOD testing. To this reviewer, the whole point of OOD testing should be to estimate performance on *unknown* but different distribution to training. In practice, this is approximated by a differently distributed test set, but there is more than a single way to have OOD samples. Here, the paper studies OOD in terms of color distribution shift but it could have other ways to do OOD. It is totally okay to use unlabelled data from *different* distributions from training, but it is not okay to (only) use the one that is literally drawn from the same distribution as the test. That is not just OOD, but rather one specific distribution, that we use to measure OOD under a controlled setting. To properly validate the claims against the proposed technique, the work should focus on creating/sampling images from all possible distribution shifts. To me, the current setup invalidates the main claims of the paper that the use of unlabeled/pseudo-labeled data can improve OOD generalization. See more details below on "requested changes".

The same parallels can be said for model selection "The high-level idea is to maximize the distribution coverage on the unlabeled dataset." is currently more like "...coverage on unlabeled dataset sampled from one specific distribution, which also happens to be the same as our test set", which is problematic in the same way as above!

---

> ### Author Response · Authors · 2022-12-13
> **Response to Reviewer h5vg**
>
> Thank you for the constructive feedback! Your review has helped us extend and refine the paper. We provide responses below.
>
> ------
>
> Request #1: OOD setting with unknown test distribution.
>
> Response: Thank you for bringing up this point, which we fully agree. Based on your suggestion, we have updated the paper and adjusted our claims to clarify that we mainly focus on the case of leveraging unlabeled data from a known test distribution to improve performance of that specific distribution shift.
>
> In addition, we include additional experiments testing on data from an unknown test distribution, where the unlabeled data is from an expanded set of distribution including the test distribution, but not solely consisting of samples that follow the test set distribution, as suggested. In this setup, the train dataset is from Clevr CoGen split A, the test dataset is from Clevr CoGen split B, and the unlabeled dataset is from the full Clevr split consisting of both split A and B with all color-shape attributes. We report results on this new setting (UNK experiments in Section 4.2) for all three base models — FiLM, TbD-net, and NS-CL. For each base model, we show comparisons against the two baselines pretrained on the new set of unlabeled data (VAE & Simclr) as well as our two DisCo sampling strategies derived from the new set of unlabeled data (DisCo-S and DisCo-G). The results are below:
>
> |                               | 0.5% Ref UNK    |
> | -----------                   | ----------- |
> | FiLM                          | 0.9265      |
> | FiLM + VAE                    | 0.9396      |
> | FiLM + SimCLR                 | 0.9255      |
> | FiLM + DisCo-S                | **0.9609**  |
> | FiLM + DisCo-G                | **0.9524**  |
>
> |                               | 0.5% Ref UNK    |
> | -----------                   | ----------- |
> | TbD-net                          | 0.8993      |
> | TbD-net + VAE                    | 0.9027      |
> | TbD-net + SimCLR                 | 0.9054      |
> | TbD-net + DisCo-S                | **0.9205**  |
> | TbD-net + DisCo-G                | **0.9156**  |
>
> |                               | 0.5% Ref UNK    |
> | -----------                   | ----------- |
> | NS-CL                          | 0.7622      |
> | NS-CL + VAE                    | 0.7583      |
> | NS-CL + SimCLR                 | 0.7727      |
> | NS-CL + DisCo-S                | **0.8016**  |
> | NS-CL + DisCo-G                | **0.7798**  |
>
> We see that DisCo reliably outperforms baselines in a setting with unknown test distribution. We have updated the main text with these descriptions and experiments in Section 4.2.

---

> > ### Author Response · Authors · 2023-01-03
> > **Looking forward to your response**
> >
> > Thank you for your review. We hope our response has clarified your concerns. Please don’t hesitate to let us know if you have any further questions!

---

### Review · Reviewer_5bCf · 2022-12-01

**Summary Of Contributions:**

This work aims to improve the compositional generalization of visual question answering (VQA) models by leveraging unlabeled, out-of-distribution image samples. This work, DisCo, proposes two techniques for better exploitation of unlabeled images: 1) an entropy-based metric for filtering pseudo-labels on unlabeled images that are not appropriate for self-training, and 2) a distribution coverage metric for selecting a model checkpoint that approximately minimizes generalization errors. Both of the techniques are based on empirical observations. The proposed learning paradigm DisCo successfully improves various VQA models (including FiLM, TbD-net, and NS-CL) on the CLEVR positional generalization dataset, and outperforms unsupervised representation baselines such as VAE and SimCLR.

**Audience:**

Yes

**Broader Impact Concerns:**

No concerns.

**Claims And Evidence:**

No

**Requested Changes:**

Additional explanation and experimental results can help to address the weaknesses as detailed above. In addition to those, the following minor changes can be made:

- The “out-of-distribution images” introduced in this work contain objects with different **compositions** of attributes, but these individual attributes (e.g., colors or shapes) have already been seen in the labeled data. It might be better to clearly explain when this work first introduces out-of-distribution images.

- In the introduction, it is mentioned that “our framework is model-agnostic and can be used for a variety of vision domains that have combinatorial structures.” Could the authors provide some specific examples other than VQA?

- Figure 2: To be consistent with the notations in Section 3.2, please use subscripts for $+$ and $-$ signs.

- Figure 3: To be consistent with the index of this whole figure, please rename the sub-figures as 3a) and 3b). Similar issue in Figure 4.

- Figure 4-left: Why do some checkpoints (points on the left side) have validation accuracy close to 0.0, while having test accuracy close to 1.0? In the text, it is mentioned that “validation set accuracies are mostly close to 1.0.” Please clarify.

**Strengths And Weaknesses:**

Strengths:

- This work studies a novel setting where the VQA model can have access to unlabeled images and extract learning signals from them. Different from prior work, this new setting explores how to achieve better compositional generalization using unlabeled data resources. Considering the efforts in collecting human annotations for the VQA task, this unsupervised learning paradigm is practical and helpful for future research.

- This work has conducted extensive experiments to validate the proposed method DisCo. Three base VQA models (FiLM, TbD-net, and NS-CL), two baselines (VAE and SimCLR), and various dataset settings have been evaluated and compared. DisCo can consistently improve the base VQA models.

- The writing is clear and easy to follow.

Weaknesses:

- This work claims to improve the compositional generalization of VQA models, which means that the model can better generalize to images that contain objects with new compositions of attributes (e.g., shape+color in this work). However, this claim does not seem to align well with the current training and evaluation protocol: At training time, unlabeled **test images** are exposed to DisCo (and other unsupervised learning baselines as well), and thus the model can learn from the different composition distribution in the test images, even though these images are not labeled. During evaluation, the model is tested with exactly the same test images or new images drawn from the same test distribution (“Ref unseen” in the experiments). In other words, the model has already learned from the test-time data distribution during training before testing. This experiment setup seems to be weak for supporting the claim of improved generalizability. In practice, it is not likely to know the test-time compositions while training the VQA model. A stronger setting might be to evaluate the learned VQA model with another distribution that is different from both the labeled and unlabeled images.

- The entropy-based metric for thresholding pseudo-labels is highly related to prior work in out-of-distribution (OOD) detection, such as [1][2][3]. A comprehensive comparison between the proposed entropy-based metric and the related methods is necessary. Moreover, on this specific task of VQA, is the proposed metric better than well-established OOD detection methods like [3]? Further experiments that compare the proposed metric and previous OOD detection methods might be helpful.

- This work is mainly established on empirical observations. Some theoretical insights could further strengthen the proposed strategies.

- The performance gain seems limited. For example, as shown in Table 1, the performance improvement brought by DisCo (from 0.7589 to 0.7760) is less than simply introducing only 0.1% labeled data from the target distribution (from 0.7589 to 0.7931). More clarification might be helpful for readers to understand how significant this improvement is.

References:

[1] Dan Hendrycks, and Kevin Gimpel. A baseline for detecting misclassified and out-of-distribution examples in neural networks. In ICLR, 2017.

[2] Shiyu Liang, Yixuan Li, and R. Srikant. Enhancing The Reliability of Out-of-distribution Image Detection in Neural Networks. In ICLR, 2018.

[3] Weitang Liu, Xiaoyun Wang, John Owens, and Yixuan Li. Energy-based out-of-distribution detection. In NeurIPS, 2020.

---

> ### Author Response · Authors · 2022-12-13
> **Response to Reviewer 5bCf**
>
> Thank you for the constructive feedback! Your review has helped us extend and refine the paper. We provide responses below.
>
> --------
>
> Weakness #1: OOD setting with unknown test distribution.
>
> Response: Thank you for bringing up this point, which we fully agree. Based on your suggestion, we have updated the paper and adjusted our claims to clarify that we mainly focus on the case of leveraging unlabeled data from a known test distribution to improve performance of that specific distribution shift.
>
> In addition, we include additional experiments testing on data from an unknown test distribution (different from both the labeled and unlabeled images), where the unlabeled data is from an expanded set of distribution including the test distribution, but not solely consisting of samples that follow the test set distribution, as suggested. In this setup, the train dataset is from Clevr CoGen split A, the test dataset is from Clevr CoGen split B, and the unlabeled dataset is from the full Clevr split consisting of both split A and B with all color-shape attributes. We report results on this new setting (UNK experiments in Section 4.2) for all three base models — FiLM, TbD-net, and NS-CL. For each base model, we show comparisons against the two baselines pretrained on the new set of unlabeled data (VAE & Simclr) as well as our two DisCo sampling strategies derived from the new set of unlabeled data (DisCo-S and DisCo-G). The results are below:
>
> |                               | 0.5% Ref UNK    |
> | -----------                   | ----------- |
> | FiLM                          | 0.9265      |
> | FiLM + VAE                    | 0.9396      |
> | FiLM + SimCLR                 | 0.9255      |
> | FiLM + DisCo-S                | **0.9609**  |
> | FiLM + DisCo-G                | **0.9524**  |
>
> |                               | 0.5% Ref UNK    |
> | -----------                   | ----------- |
> | TbD-net                          | 0.8993      |
> | TbD-net + VAE                    | 0.9027      |
> | TbD-net + SimCLR                 | 0.9054      |
> | TbD-net + DisCo-S                | **0.9205**  |
> | TbD-net + DisCo-G                | **0.9156**  |
>
> |                               | 0.5% Ref UNK    |
> | -----------                   | ----------- |
> | NS-CL                          | 0.7622      |
> | NS-CL + VAE                    | 0.7583      |
> | NS-CL + SimCLR                 | 0.7727      |
> | NS-CL + DisCo-S                | **0.8016**  |
> | NS-CL + DisCo-G                | **0.7798**  |
>
> We see that DisCo reliably outperforms baselines in a setting with unknown test distribution. We have updated the main text with these descriptions and experiments in Section 4.2.
>
>
> --------
>
> Weakness #2: Entropy-based metric comparisons.
>
> Response: Thank you for your suggestions. We have cited these papers, and discuss them in the updated paper. Specifically, compared to prior work, we explore the task of compositional generalization, instead of the general OOD setting. We also show that an entropy-based metric can improve such generalization performance for the complex question answering task, instead of detecting OOD examples in the classification task [1, 2, 3]. Notably, our paper has employed a simple entropy-metric that is easy to compute and can be easily implemented across model architectures without fine-tuning (c.f. [3]). We agree that other metrics can also be used, such as variants of the softmax score [1, 2] and the energy score [3], which may be fruitful to explore as future work. We have updated the main text to clarify this point in the limitations (Section 4.4).
>
> [1] ​​Hendrycks, D., & Gimpel, K. (2016). A Baseline for Detecting Misclassified and Out-of-distribution Examples in Neural Networks. ICLR, 2017.
>
> [2] Liang, S., Li, Y., & Srikant, R. (2017). Enhancing the Reliability of Out-of-distribution Image Detection in Neural Networks. ICLR, 2018.
>
> [3] Liu, W., Wang, X., Owens, J., & Li, Y. (2020). Energy-based Out-of-distribution Detection. Advances in Neural Information Processing Systems, 33, 21464-21475.
>
> --------
>
> Weakness #3: Theoretical insights.
>
> Response: Our paper focuses on utilizing strong empirical evidence for compositional generalization performance. We believe that these insights are interesting to the community, and welcome theoretical insights for compositional generalization as open questions for future work.
>
> --------
>
> Weakness #4: Performance gain in the fully biased setting.
>
> Response: The reviewer notes that the performance improvement brought by DisCo for the fully biased case is less than introducing 0.1% labeled data. We note that in many scenarios, it is useful to perform well without any labeled data in the compositional generalization setting, e.g., when we only have access to objects with a specified set of attributes during training and do not know the test distribution, or when we need to adapt to new test distributions without labels. We have added this clarification in the main text (Section 4.2).

---

> > ### Author Response · Authors · 2022-12-13
> > **Response to Reviewer 5bCf (continued)**
> >
> > Request #1: Compositional generalization and out-of-distribution images.
> >
> > Response: Thank you for the feedback. We have clarified in the main text (Section 1) that we are interested in the task of compositional generalization, where we focus on the distribution shift of combinations of object attributes, while having seen each individual attribute in the training set under different combinations.
> >
> > --------
> >
> > Request #2: Additional examples for vision domains with combinatorial structures.
> >
> > Response: Additional examples of vision domains with combinatorial structures can be found in the tasks of referring expression comprehension, grounded instructions, robotic manipulation, etc (for example, in datasets such as ReferIt3D [4], Alfred [5], BabyAI [6],  CLIPort [7]). We have added these examples in the main text (Section 1).
> >
> > [4] Achlioptas, P., Abdelreheem, A., Xia, F., Elhoseiny, M., & Guibas, L. (2020, August). Referit3D: Neural Listeners for Fine-grained 3D Object Identification in Real-world Scenes. In European Conference on Computer Vision (pp. 422-440). Springer, Cham.
> >
> > [5] Shridhar, M., Thomason, J., Gordon, D., Bisk, Y., Han, W., Mottaghi, R., ... & Fox, D. (2020). Alfred: A Benchmark for Interpreting Grounded Instructions for Everyday Tasks. In Proceedings of the IEEE/CVF conference on computer vision and pattern recognition (pp. 10740-10749).
> >
> > [6] Chevalier-Boisvert, M., Bahdanau, D., Lahlou, S., Willems, L., Saharia, C., Nguyen, T. H., & Bengio, Y. (2018). Babyai: A Platform to Study the Sample Efficiency of Grounded Language Learning. arXiv preprint arXiv:1810.08272.
> >
> > [7] Shridhar, M., Manuelli, L., & Fox, D. (2022, January). Cliport: What and Where Pathways for Robotic Manipulation. In Conference on Robot Learning (pp. 894-906). PMLR.
> >
> > --------
> >
> > Request #3 & #4: Figure notations.
> >
> > Response: Thank you for pointing out these details. We have updated the figures accordingly.
> >
> > --------
> >
> > Request #5: Validation accuracy in Figure 4-left.
> >
> > Response: We apologize for making an error in parsing the validation accuracy, and have updated the figure accordingly with the correct results. We deeply thank you for pointing out this error. The new plot shows the same trend and conclusion, with validation set accuracies mostly close to 1.0 and varied test set accuracy, resulting in low correlation and poor model selection. We have also updated the Pearson’s correlation for this plot to be 0.2565, compared with DisCo’s at 0.6066.

---

> > > ### Comment · Reviewer_5bCf · 2022-12-20
> > > **Follow-up Question for Authors: OOD Setting with Unknown Test Distribution**
> > >
> > > Thank you for your response, which has addressed most of my previous concerns. In particular, the adjusted claims and the evaluation on a new test distribution is greatly appreciated. However, one more follow-up question remains:
> > >
> > > In the initial setting, DisCo is evaluated by training on **labeled Clevr CoGen split A + unlabeled Clevr CoGen split B**, and then tested on Clevr CoGen split B. As suggested by all the reviewers, this setting is not realistic since the specific test distribution should be unknown at training time. The authors have adjusted the claims and re-evaluated DisCo by training on **labeled Clevr CoGen split A + unlabeled full Clevr CoGen split**, and then tested on Clevr CoGen split B. Since the full split contains data in split B, this setup is still weak for demonstrating improved compositional generalization.
> > >
> > > To further strengthen the revised experiments, the authors might consider some setup similar to either of the following:
> > > - Train on labeled split A + unlabeled split C (different from A and B), test on split B
> > > - Train on labeled split A + unlabeled split B, test on full split
> > >
> > > It would be great to show that by learning from unlabeled data from another composition distribution, the model can generalize better to other different composition distributions which is totally unseen during training.

---

> > > > ### Author Response · Authors · 2022-12-23
> > > > **Follow-up Response to Reviewer 5bCf**
> > > >
> > > > Thank you for your reply. We are glad that our response has addressed most of your previous concerns. Based on your suggestion, we ran additional experiments on the proposed setup — train on labeled split A and unlabeled split B, and tested on the full split. The results for DisCo on each of the three base models are below, showing consistent performance gain compared to base and baseline models. We have added these new experimental results to the Appendix (Section A.6).
> > > >
> > > > |                               | 0.5% Ref UNK    |
> > > > | -----------                   | ----------- |
> > > > | FiLM                          | 0.9634     |
> > > > | FiLM + VAE                    | 0.9683      |
> > > > | FiLM + SimCLR                 | 0.9635      |
> > > > | FiLM + DisCo-S                | **0.9798**  |
> > > > | FiLM + DisCo-G                | **0.9745**  |
> > > >
> > > > |                               | 0.5% Ref UNK    |
> > > > | -----------                   | ----------- |
> > > > | TbD-net                          | 0.9506      |
> > > > | TbD-net + VAE                    | 0.9495      |
> > > > | TbD-net + SimCLR                 | 0.9510      |
> > > > | TbD-net + DisCo-S                | **0.9582**  |
> > > > | TbD-net + DisCo-G                | **0.9536**  |
> > > >
> > > > |                               | 0.5% Ref UNK    |
> > > > | -----------                   | ----------- |
> > > > | NS-CL                          | 0.8810      |
> > > > | NS-CL + VAE                    | 0.8789      |
> > > > | NS-CL + SimCLR                 | 0.8872      |
> > > > | NS-CL + DisCo-S                | **0.9011**  |
> > > > | NS-CL + DisCo-G                | **0.8902**  |
> > > >
> > > > To summarize, the original experiments, together with the two added experiments during the rebuttal, have covered three settings. Following your notation, we name the training and test distributions as the following: train on labeled split A + unlabeled split C, test on split B.
> > > >
> > > > The three settings are:
> > > >
> > > > 1) Original paper setting: A = CoGenA, C = CoGenB, B = CoGenB (B = C.)
> > > >
> > > > 2) Added experiments in our first response: A = CoGenA, C = CoGenA+CoGenB, B = CoGenB (B is a subset of C.)
> > > >
> > > > 3) Added experiments in this response as suggested: A = CoGenA, C = CoGenB, B = CoGenA+CoGenB (B is a superset of C.)
> > > >
> > > > Our proposed method shows improvements over baselines across all three settings, although the last setting is not the main focus of this paper. We clarify that our paper focuses on a compositional generalization setting where unlabeled data covers the test distribution (though the actual distribution may not be exactly the same). We believe that our initial experiments as well as follow-up experiments have demonstrated this claim, and we have edited the paper to accentuate this point.
> > > >
> > > > We hope this clarifies the reviewer’s concerns. Please don’t hesitate to let us know if you have any further questions.

---

> > > > > ### Comment · Reviewer_5bCf · 2022-12-25
> > > > > **Thank you for the response**
> > > > >
> > > > > Thank the authors for the response. The newest reply has addressed my remaining concern with experimental results.

---

> > > > > > ### Author Response · Authors · 2022-12-27
> > > > > > **Thanks to the reviewer**
> > > > > >
> > > > > > Thank you for your time and feedback on the paper!

---

### Decision · Action_Editors · 2023-01-04

**Recommendation:** Accept as is

**Comment:**

This paper proposes a learning method called DisCo for better distribution coverage to improve compositional generalization for visual reasoning on the CLEVR dataset. After author response, it received 2 Leaning Accept, and 1 Accept recommendations.

On one hand, all the reviewers agree that the paper is clearly written, the proposed method is reasonable and the results support the core claim of compositional generalization after revision. The experiments are thorough and the ablation is clear.

On the other hand, the main concern raised by all the reviewers is that the experimental setup does not fully support the broad claim of compositional generalization. Specifically, the model has already learned from the test-time data distribution during training before testing. This experiment setup is a little bit weak, since the specific test distribution should be unknown at training time. During rebuttal, the authors have adjusted the claims and added some new results, which the reviewers appreciated.

Another weakness is that all the experiments are performed on the toy CLEVR dataset, so it remains unclear how the technique developed in this paper is really useful for real-world datasets. The authors have added this limitation discussion in the revision.

Overall, the rebuttal solved most of the concerns raised by the reviewers, and all the reviewers tend to accept the paper. Though some weaknesses still exist, the paper itself is still self-contained, and will be interesting to researchers specifically working in this sub-field. Therefore, the editor decided to recommend accept by the end.

**Audience:**

This paper works on improving the compositional generalization for visual reasoning and specifically using the CLEVR dataset for experiments. A subset of the TMLR audience will be interested in this paper.

**Claims And Evidence:**

The claims in the initial draft seem to be boarder than the evidence presented in the paper. During the author response, the authors have adjusted the claims, and added some new results, which the reviewers feel satisfied with.